# Structuring Representations Using Group Invariants

**Mehran Shakerinava**[†]**, Arnab Kumar Mondal**[†]**, Siamak Ravanbakhsh**
McGill University and Mila, Montréal, Canada
{mehran.shakerinava, arnab.mondal, siamak.ravanbakhsh}@mila.quebec

## Abstract

A finite set of invariants can identify many interesting transformation groups. For example, distances, inner products and angles are preserved by Euclidean, Orthogonal and Conformal transformations, respectively. In an equivariant representation, the group invariants should remain constant on the embedding as we transform the input. This gives a procedure for learning equivariant representations without knowing the possibly nonlinear action of the group in the input space. Rather than enforcing such hard invariance constraints on the latent space, we show how to use invariants for "symmetry regularization" of the latent while guaranteeing equivariance through other means. We also show the feasibility of learning disentangled representations using this approach and provide favorable qualitative and quantitative results on downstream tasks, including world modeling and reinforcement learning.

## 1 Introduction

Sample efficient representation learning is a critical open challenge in deep learning for AI. When we have prior information about transformations that are relevant to a particular domain, building representations that are aware of these transformations can lead to better sample efficiency and generalization. One way to use such symmetry priors is to make the network invariant to the given transformations. A generalization of this idea is called equivariance, where transforming the input transforms the output in a specific way. An equivariant network that makes good predictions for a particular input also generalizes to all input transformations, making symmetry a useful prior.

While recent years have witnessed a range of exciting equivariant deep models, there are several limitations. First, most equivariant networks constrain the network architecture, often requiring specialized implementations. Moreover, transformations considered by the existing methods are often assumed to be linear in both input and representation space. This is the case for architectures designed for finite permutation groups and continuous Lie groups. Approaches that go beyond linear transformations in the input space often assume access to group information – *i.e.*, the group member that transforms one input to another is known. This paper introduces a simple approach that addresses all of these limitations.

Our approach uses the invariants of a given linear representation of a transformation group. Previously invariants were used to connect different geometries, and group theory in Klein's Erlangen program [32]. According to this view, geometries are concerned with invariant quantities under certain transformations. For example, Euclidean geometry is concerned with the length, angle, and parallelism of lines, among others, because Euclidean transformations preserve these. However, moving to the more general and less structured Affine geometry, notions of distance and angle are no longer relevant, while parallelism remains an invariant of the geometry. The corresponding symmetry groups are examples of Lie groups that have a subgroup relation, $E(n) < Aff(n)$, thereby enabling the groups to characterize a hierarchy (or lattice) of different geometries.

---

[†]These authors contributed equally to this work.

36th Conference on Neural Information Processing Systems (NeurIPS 2022).

From this geometric perspective, our proposal in this work is to induce a geometry on the embedding and make it equivariant to a given group by enforcing the invariants of their defining action. For example, distance is the invariant for Euclidean geometry, which means all distance-preserving transformations are Euclidean. Therefore, to enforce equivariance to the Euclidean group, it is sufficient to ensure that the embedding of any two data points has the same distance before and after the same transformation of the inputs; see Figure 1. While this approach uses the *defining action* of different groups in the embedding space, the same group can have a non-linear and unknown action on the input space. In the pendulum example of Figure 1, the group $E(3)$ acts on the value of each input image pixel using an unknown and non-linear action. Moreover, this approach does not require the pairing of group members with transformations, a piece of information that is often unavailable.

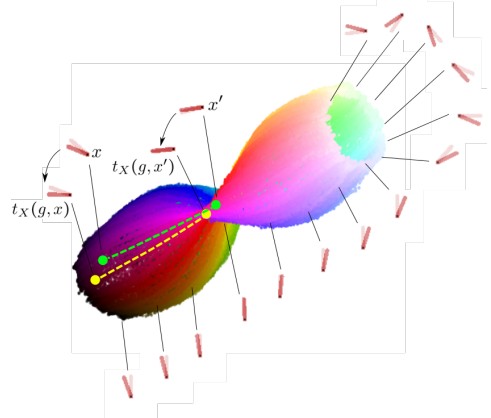

Figure 1: **$E(3)$-equivariant embedding for the pendulum.** The input $x$ consists of a pair of images that identify both the angle and the angular velocity of a pendulum. The equivariant embedding learns to encode both: the true angle is shown by a change of color and angular velocity using a change of brightness. The two circular ends (black and white) correspond to states of maximum angular velocity in opposite directions. The SymReg objective for the *Euclidean group* learns this embedding by preserving the pairwise distance between the codes before $(f(x), f(x'))$ and after $(f(t_X(g,x)), f(t_X(g,x)))$ transformations of the input by $t_X$. Therefore dashed lines have equal lengths. For the pendulum, the transformations are in the form of applying positive or negative torque in some range.

In the rest of the paper, we arrive at the idea above from a different path: after reviewing related works in Section 2 and providing a background in Section 3, Section 4 observes that equivariance, in its general form, can be a weak inductive bias. This is because having an injective code is sufficient for equivariance to "any" transformation group. However, in this manifestation of equivariance, the group action on the embedding can be highly non-linear. Since the simplicity of the action on the embedding seems essential for equivariance to become a useful learning bias, Section 5 proposes to *regularize* the group action on the code to make it "simple". This *symmetry regularization* (SymReg) objective is group-dependent and the essence of our approach. Enforcing geometric invariants in the latent space is proposed as a symmetry regularization. While we focus on equivariant representation learning through self-supervision, in principle, supervised tasks can also benefit from the proposed SymReg. An important benefit of a symmetry-based representation is its ability to produce disentangled representations through group decomposition [27]. Section 6 studies disentanglement using SymReg. Section 7 presents a range of experiments to understand its behavior and puts it in the context of comparable baselines.

## 2  Related Works

Finding effective priors and objectives for deep representation learning is an integral part of the quest for AI [3]. Among these priors, learning equivariant deep representations has been the subject of many works over the past decade. Many recent efforts in this direction have focused on the design of equivariant maps [57, 13, 47, 34, 15, 23, 54, 19, 6] where the "linear" action of the group on the data is known. A particularly relevant example here is Villar et al. [54], which uses group invariants to construct equivariant maps where the group acts using its linear defining action in the input space. Due to this constraint, the application of these models has been focused on fixed geometric data such as images [36], sets [60, 45], graphs [39, 33], spherical data and the (special) orthogonal group [14, 1, 50, 22], the Euclidean group [52, 55, 24] or other physically motivated groups such as the Lorentz [4] or Poincare group [54], among others.

In the present work, the group action is unknown and possibly non-linear. Our setup is closer to the body of work on generative representation learning [7, 11, 40], in which the (linear) transformation is applied to the latent space [46, 58, 35, 37, 16, 21]. Among these generative coding methods, transforming autoencoder [29] is a closely related early work, which in addition to equivariance,

seeks to represent the part-whole hierarchy in the data. What additionally contrasts our work with the follow-up works on capsule networks [48, 38] is that SymReg is agnostic to the choice of architecture and training. We only rely on our objective function to enforce equivariance.

Since we consider learning equivariant representations through self-supervision, exciting recent progress in this area is also quite relevant [25, 42, 9, 53, 26, 61, 20, 41]. While the use of transformations is prominent in these works, in many settings, the objective encourages *invariance* to certain transformations, making such models useful for invariant downstream tasks such as classification. Similar to many of these methods, we also use transformed pairs to learn a representation, with the distinction of learning an *equivariant* representation. An exception is the recent work of Dangovski et al. [17], which learns an equivariant representation by separating the invariant embedding from the pose, where the relative pose is learned through supervision. Therefore, in that work, in contrast to ours, one needs to know the transformation that maps one input to another. When considering the Euclidean group, SymReg preserves distances in the embedding space under non-linear transformations of the input. This embedding should not be confused with isometric embedding [51], where the objective is to maintain the pairwise distances between points in the input and the embedding space.

## 3 Background on Symmetry Transformations

We can think of transformations as a set of bijective maps on a domain $X$. Since these maps are composable, we can identify their compositional structure using an abstract group $G$. For this reason, such transformations are called group actions. To formally define transformation groups, we first define an abstract group. A *group* $G$ is a set equipped with a binary operation, such that the set is closed under the operation $gg' \in G \; \forall g, g' \in G$, every $g \in G$ has a unique inverse such that $gg^{-1} = e$, where $e$ is the identity element of the group, and the group operations are associative $(gg')g'' = g(g'g'')$.

A *$G$-action* on a set $X$ is defined by a function $t : G \times X \to X$, which can be thought of as a bijective transformation parameterized by $g \in G$. In order to maintain the group structure, the action should satisfy the following two properties: (1) the action of the identity is the identity transformation $t(e, x) = x$; (2) the composition of two actions is equal to the action of the composition of group elements $t(g, t(g'x)) = t(gg', x)$. The action $t$ is *faithful* to $G$ if transformations of $X$ using each $g \in G$ are unique – *i.e.*, $\forall g, g' \; \exists x \in X$ s.t. $t(g, x) \neq t(g', x)$. If a $G$-action is defined on a set $X$, we call $X$ a $G$-set. Many groups are defined using their defining action; for example, $SO(3)$ is the group of rotations in 3D space. While this defining action is a linear transformation, the same group can act non-linearly on $\mathbb{R}^n$ using the action $t : SO(3) \times \mathbb{R}^n \to \mathbb{R}^n$.

## 4 Equivariance is Cheap, Actions Matter

A symmetry-based representation or embedding is a function $f : X \to Z$ such that both $X$ and $Z$ are $G$-sets, and furthermore, $f$ "knows about" $G$-actions, in the sense that transformations of the input using $t_X$ have the same effect as transformations of the output using some action $t_Z$:

$$f(t_X(g, x)) = t_Z(g, f(x)) \quad \forall g, x \in G \times X \tag{1}$$

The following claim shows that despite many efforts in designing equivariant networks, simply asking for the representation to be equivariant is not a strong inductive bias, and we argue that the action matters. Put another way, the strong performance of existing equivariant networks should be attributed to the fact that the group action on the embedding space is simple (linear).

**Proposition 4.1.** *Given a transformation group $t_X : G \times X \to X$, the function $f : X \to Z$ is an equivariant representation if $\forall g \in G, x, x' \in X$*

$$f(x) = f(x') \Leftrightarrow f(t_X(g, x)) = f(t_X(g, x')). \tag{2}$$

*That is, two embeddings are identical iff they are identical for all transformations.*

The proof is in the appendix. The condition above is satisfied by all injective functions, indicating that many functions are equivariant to any group.

**Corollary 4.2.** *Any* injective *function $f : X \to Z$ is equivariant to any transformation group $t_X : G \times X \to X$, if we define $G$ action on the embedding space as*

$$t_Z(g, z) \doteq f(t_X(g, f^{-1}(z))) \quad \forall g, z \in G \times Z \tag{3}$$

The ramification of the results above in what follows is two-fold:

**1.** While injectivity ensures equivariance, the group action on the embedding, as shown in Equation (3), can become highly non-linear. Intuitively, this action recovers $x = f^{-1}(z)$, applies the group action $x' = t_X(x)$ in the input domains and maps back to the embedding space $f(x')$ to ensure equivariance. In the following, we push $t_Z$ towards a simple linear $G$-action through optimization of $f$. This objective can be interpreted as a *symmetry regularization or a symmetry prior* (SymReg).

**2.** Although Corollary 4.2 uses injectivity of $f$ for the entire $X$, we only need this for the data manifold. In practice, one could enforce injectivity on the training dataset $D$ using a decoder, architectural choices such as momentum encoder [26], or loss functions defined on the training data, such as a hinge loss [25] $L_{\text{hinge}}(f, D) = \sum_{x, x' \neq x \in D} \max(\epsilon - \|f(x) - f(x')\|, 0)$ or other losses that monotonically decrease with distance, such as $\frac{1}{\|f(x) - f(x')\|}$, or its logarithm $-\log(\|f(x) - f(x')\|)$. In experiments, we use the logarithmic barrier function.

## 5 Symmetry Regularization Objectives

In learning equivariant representations, we often do not know the abstract group $G$ and how it transforms our data, $t_X$. We assume that one can pick a reasonable abstract group $G$ that "contains" the ground truth abstract group acting on the data – *i.e.*, $G$ action on the input may not be faithful. Our goal is to learn an $f : X \to Z$ that is equivariant w.r.t. the actions $t_X, t_Z$, where $t_X : G \times X \to X$ is unknown and $t_Z$ is some (simple) $G$-action on $Z$ of our choosing.

**More Informed but Less Practical Setting.** In the most informed case, the dataset also contains information about which group member $g \in G$ can be used to transform $x$ to $x'$ – that is, the dataset consists of triples $(x, g, x_t = t_X(g, x))$. By having access to this information, we can regularize the embedding using the following loss function: $L_G^{\text{informed}}(f, D) = \sum_{(x, g, x_t) \in D} \ell(f(x_t) - t_Z(g, f(x)))$, where $\ell$ is an appropriate loss function, such as the square loss. At its minimum, we have $f(x_t) = t_Z(g, f(x))$ or $f(t_X(g, x)) = t_Z(g, f(x))$, enforcing equivariance condition of Equation (1). However, even if the optimal value is not reached, due to its injectivity, $f$ is still $G$-equivariant, and the the objective above is regularizing the $G$ action on the code. This informed setup is used in equivariant contrastive learning of [17]. The assumption of having access to $g$ is realistic when we know the action $t_X$, so that we can generate $(x, g, x_t)$ triplets. Fortunately, using group invariants, we may still learn an equivariant embedding, even if we do not have the group information tied to the dataset.

Here, we introduce our method for several well-known groups first and then elaborate on the more general treatment.

*Example* 1 (**Euclidean Group**). The defining action of the Euclidean group $E(n)$ is the set of transformations that preserve the Euclidean distance between any two points in $\mathbb{R}^n$, a.k.a. isometries. These transformations are compositions of translations, rotations, and reflections. Since, for the real domain, all Euclidean isometries are linear and belong to $E(n)$, we can enforce the group structure on the embedding by ensuring that distances between the embeddings before and after any transformation match. For this, we need the dataset $D$ to be a set of pairs of pairs $\big((x, x_t = t_X(g, x)), (x', x'_t = t_X(g, x'))\big)$, where $x, x'$ are transformed using the same *unknown* group member $g$. Distance-preservation loss below combined with injection loss are sufficient to produce an $E(n)$-regularized embedding:

$$L_{E(n)}(f, D) = \sum_{\big((x, x_t), (x', x'_t)\big) \in D} \ell\big(\overbrace{\|f(x) - f(x')\|}^{\substack{\text{distance before the} \\ \text{transformation}}} - \overbrace{\|f(x_t) - f(x'_t)\|}^{\substack{\text{distance after the} \\ \text{transformation}}}\big) \tag{4}$$

For example, in the standard RL setup, where we have access to triplets $(s, a, s')$, we can easily form $D$ by unrolling an episode and collecting two different state transitions corresponding to a particular action. In practice, with a finite number of actions, we can efficiently generate this dataset by keeping a separate buffer for each action where we store state transitions for that action and sample from that buffer to train the embedding function $f$. We provide the algorithm in Appendix C.

*Example* 2 (**Orthogonal and Unitary Groups**). The defining action of the orthogonal group $O(n)$ preserves the inner product between two vectors. The analogous group in the complex domain is the

unitary group, which preserves the complex inner product. Our symmetry-regularization objective enforces this invariant: $L_{O(n)}(f, D) = \sum\limits_{\left((x,x_t),(x',x'_t)\right)\in D} \ell\left(f(x)^\top f(x') - f(x_t)^\top f(x'_t)\right).$

For the unitary group, one additionally needs to embed to complex domain $Z = \mathbb{C}^n$, where the only difference is in the definition of the inner product.

*Example* 3 (**Conformal Group**). The invariant of conformal geometry is the angle. In a Euclidean embedding, conformal transformations include a combination of translation, rotation, dilation, and inversion with respect to an $n-1$-sphere. To enforce this group structure, we need triplets of inputs before and after a transformation $\left((x, x_t), (x', x'_t), (x'', x''_t)\right)$, so that we can calculate the angle in the embedding. Conformal SymReg objective, which preserves angles, imposes a weaker constraint on the embedding than the distance preservation of the Euclidean group – since the latter implies the former. Moreover, it has an additional benefit that, compared to $L_{E(n)}$, the loss cannot be minimized by simply shrinking the embedding. Therefore in practice, the injection enforcing losses of Section 4 is not as crucial when using conformal symmetry regularization.

## 5.1 General Setting

Given a group $G$ acting linearly on a vector space $Z$, *invariant polynomials* associated with this action are those polynomials satisfying $P(t_Z(z, g)) = P(z) \ \forall g \in G$. These polynomials form an *algebra* studied in the field of *invariant theory* [56, 44]. In particular, a relevant problem is the question of whether there exists a finite set of bases for invariant polynomials for a given group representation. This question was one of Hilbert's 23 problems, and it was answered affirmatively by Hilbert himself for linear *reductive groups*, which includes classical Lie groups [28]. *Our proposal, in its most general form is to ensure invariance of polynomial bases within the orbits of the latent space before-after transformation of the input.*

Some examples of classical Lie groups and their invariants are: volume and orientation preservation by the Special Linear group, where the corresponding invariant polynomial is the determinant; Lorentz and Poincare groups are the analogs of the Orthogonal and Euclidean groups in the Minkowski space respectively, therefore equipped with similar invariants; the Symplectic group preserves another bilinear form. Finite groups also possess invariants. We show this use of invariants for SymReg through the important example of the symmetric group.

*Example* 4 (**Symmetric Group**). Symmetric polynomials $P(z_1, \ldots, z_n)$ that are invariant under all permutations of variables have a finite set of elementary bases:

$$e_1(z) = \sum_{1 \le j \le n} z_j, \quad e_2(z) = \sum_{1 \le j < k \le n} z_j z_k, \quad \ldots, \quad e_n(z) = z_1 z_2 \ldots z_n.$$

Assuming an n-dimensional embedding (*i.e.*, $z_j \in \mathbb{R}$), the corresponding SymReg objective penalizes change in these elementary basis before-after a transformation. At its minimum value, this penalty ensures that transformations of the input lead to permutations of the latent dimensions – however, with SymReg, this loss is used only to *regularize* the embedding. An alternative approach to SymReg for finite groups and, in particular, the Symmetric group is discussed in Appendix B.

**Choice of Lie group** Deciding on a Lie group for each application and in particular working with the corresponding invariants can be cumbersome. A simple alternative is to use an $E(n)$-equivariant embedding for sufficiently large $n$. This is because Lie groups have isometric Euclidean embedding for sufficiently large $n$. We demonstrate this in the experiments with $SO(3)$ group in Section 7.1.

## 6 Decomposing the Representation

Higgins et al. [27] suggested a notion of disentangled representation based on decomposition of the abstract group into a direct product form $G = G_1 \times \ldots \times G_k$. There are two approaches to learning such decomposed representation using SymReg, depending on whether or not we can perform certain types of transformations in isolation. For example, an RL agent may transform its environment through actions like moving a single limb that can be performed in isolation. In this case, we call the decomposition *active* to contrast it with the *passive* case, where the action of different subgroups is always mixed in our dataset.

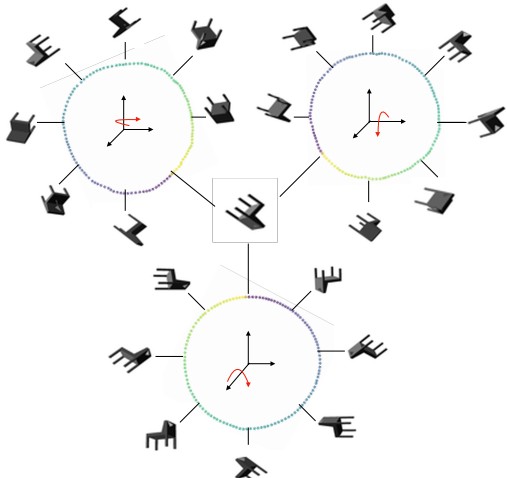

Figure 2: **Visualization of SymReg's latent projection for the rotating Chair dataset.** The chair is rotated in three orthogonal axes from 0 to $2\pi$. The latent embedding for each chair pose is projected from a 16D embedding space to a 2D space for visualization. The colors of the representations are mapped to the chair's angle of rotation. We notice that the mapping function $f$ learned is continuous with respect to the transformations of the object, and it maps the rotations along an axis to a circular manifold. This is true for each orthogonal axis of rotation. We observe a similar result for any other initial pose for the chair. A qualitative comparison with an existing method [25] is provided in Appendix D.1.

**Active Decomposition.** Let $G = \{(g_1, \ldots, g_k) \in G_1 \times \ldots \times G_k\}$ be a product group, where $G_i \cong \{(e, \ldots, e, g_i, e, \ldots, e) \in G\}$ can be identified with a normal subgroup of $G$. In active decomposition, sub-groups can act in isolation, and therefore we have $k$ types of tuples in our dataset $D_1, \ldots, D_k \subset D$. Each subset $D_i$ is associated with actions of a subgroup $G_i$ using $t_X((e, \ldots, e, g_i, e, \ldots, e), \cdot))$, $g_i \in G_i$. In this setting, the representation $f : X \to Z = Z_1 \times \ldots \times Z_k$ can be thought of as $k$ separate functions where $f_i : X \to Z_i$ is equivariant to $G_i$-action and invariant to all $G_j, j \neq i$ actions. This gives the following objective

$$L_G^{\text{active}}(f, D) = \sum_{i=1}^{k} \underbrace{L_{G_i}(f_i, D_i)}_{\text{equivariance to } G_i} + \underbrace{L_{G/G_i}^{inv.}(f_i, D \backslash D_i)}_{\text{invariance to } G_j \text{ for } j \neq i}, \tag{5}$$

where $L_G^{inv.}(f, D)$ enforces invariance of $f$ to $G$-transformations in $D$ – e.g., by penalizing $\|f(x) - f(t_X(g, x))\|$.

**Passive Decomposition.** When we have no control over transformations, and we are simply given the data, it is still possible to use an abstract group that has a product form. Here again, $f : X \to Z = Z_1 \times \ldots \times Z_k$, but the loss function is simply enforced on each block separately – i.e., $L_G^{\text{passive}}(f, D) = \sum_{i=1}^{k} L_{G_i}(f_i, D)$, where $L_{G_i}(f_i, D)$ is a SymReg objective from Section 5.

## 7 Experiments

We conducted many experiments to qualitatively study the representation learned by SymReg and its ability to produce a disentangled representation, and quantitatively compare it against simple baselines in representation learning and downstream RL tasks. For details on architecture and training, see Appendix G.

### 7.1 Qualitative Analysis

In this section, we visualize the representation learned for two examples from the Gym environment [5], including the pendulum and the mountain car (see Appendix D.2), followed by an experiment involving a rotating object where we know the ideal embedding is the $SO(3)$ manifold. Finally, Figure 3 visualizes a conformal embedding for double-bump world. In most cases, we also visualize a Variational AutoEncoder (VAE) [30] embedding for comparison. Our objective here is to visually demonstrate the behavior of SymReg and its remarkable ability to learn an embedding informed by the non-linear transformation of the input.

**The Pendulum.** For this experiment, the input $x$ is two consecutive frames of the pendulum that have been grayscaled and downsampled to $32 \times 32$ pixels. The action space is a range of torques

that can be applied to the base of the pendulum. We use the action to transform the data. We use the objective of Equation (4) to learn an $E(3)$-equivariant representation. To efficiently estimate $L_{E(n)}$, we use a mini-batch that consists of 64 randomly sampled observations from the environment and their transformations via three randomly sampled actions ($4 \times 64$ samples in total). The model learns to parameterize the embedding using the angle and the angular momentum of the pendulum from the input data; see Figure 1. In order to compare, we visualize the learned latent of VAE and run similar experiments on the Mountain Car Environment in Appendix D.2).

**Rotating Chair.**    We consider a 3D chair from ModelNet40 [59] and transform it through the action of the group $SO(3)$. The group action on the input is the 2D projection into a $48 \times 48$ image after the 3D rotation of the chair. While the group of interest is $SO(3)$, we use SymReg loss of Equation (4) following Section 5.1. We embed the chair in $\mathbb{R}^{16}$ using SymReg and visualize the latent by rotating the chair along three orthogonal axes and projecting the latent codes into a 2D space. Figure 2 shows three circular latent traversals of SymReg embedding corresponding to rotation around each axis, which is consistent with the structure of the $SO(3)$ manifold. The process of learning the $SO(3)$ manifold is a challenging task (see Appendix D.1), and previous works assumed that the group member corresponding to each transformation is given [46, 2]. In contrast, we only use the observations corresponding to similar actions during training and not the group members themselves. As we see later, this is critical in settings such as RL, where group information is unavailable.

**Conformal Embedding for Double-Bump World.**    Double-bump world consists of a rectangular bump signal and a triangular bump signal, both cyclically shifted and superimposed. These transformations are given by a pair $(\Delta_1, \Delta_2)$ which cyclically shifts the rectangular bump by $\Delta_1$ and the triangular bump by $\Delta_2$. In our experiments, the signal length is 64, and the length of the bump is 16. SymReg em-

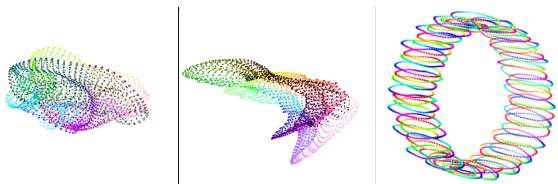

Figure 3: (left) SimCLR (middle) VAE (right) SymReg embeddings of double-bump world.

beds to a 4-dimensional conformal space in this example. Figure 3(right) shows the random projection of the embedding, where the colors change as the triangle bump moves. The figure suggests that SymReg can learn to represent a data point using the location of two bumps. For comparison, we show the embedding found by VAE and SimCLR [9].

### 7.2    Experiments on Active and Passive Decomposition

In this section, we first contrast active and passive decomposition in their ability to disentangle the two bumps in the double bump world. We observe that while both can decompose the embedding into a product form $SO(2) \times SO(2)$, only active decomposition leads to disentanglement. Finally, we apply active decomposition to the more complex setting of ego-motion, where SymReg can decompose the representation of the agent's state into location and orientation.

**Decomposition of the Double-Bump World.**    Here, we compare the active and passive decomposition for the same double-bump world. While the ground truth is $SO(2) \times SO(2)$, SymReg uses the larger group $E(2) \times E(2)$. In the active case, each subgroup moves one of the bumps, and the loss of Equation (5) is used to learn an embedding for each subgroup. In the passive case, both bumps move randomly. Figure 4 compares the decomposed embedding found in each case. While in both cases, the $SO(2) \times SO(2)$ torus is decomposed into a product of circles, only the active case successfully disentangles the two bumps. Note that the color of each point is based on the location of the triangle bump. Our results agree with Caselles-Dupré et al. [8], who claim that learning a disentangled representation requires interaction with the environment; see also [43, 40]. However, we note that while the disentangling of the bump movements does not happen in the passive case, we can still successfully "decompose" the embedding.

**Active Decomposition for Ego-Motion.**    We used a modified version of the single-room environment of MiniWorld [12] for this experiment. The agent is standing in a 3D room containing eight differently colored boxes around the walls. A map of the room can be seen in Figure 5. An

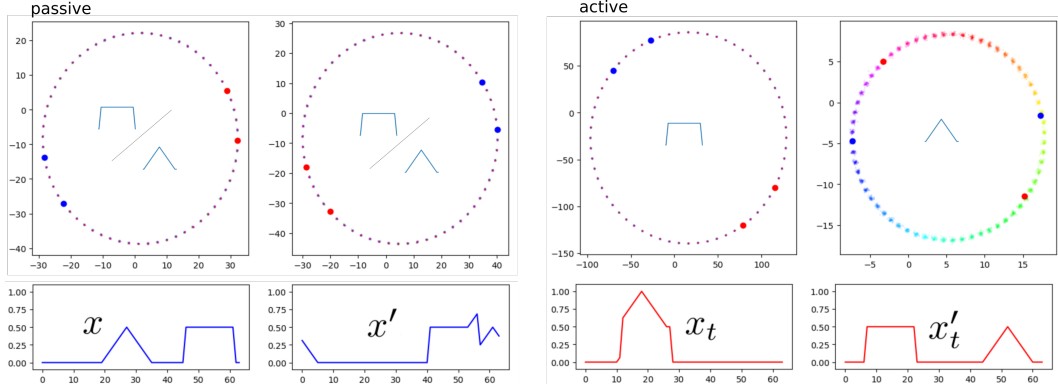

Figure 4: **Active versus passive decomposition for the double-bump world.** Four images at the bottom show a pair of inputs $(x, x')$ and their transformations $(x_t, x'_t)$. The figure shows the embedding for two inputs before, $(x, x')$ in blue, and after, $(x_t, x'_t)$ in red, the *same* transformation. This transformation cyclically shifts both the triangle and the square to the left, but the amount of translation is larger for the square. In both passive and active decomposition, the Euclidean distance is preserved by the transformation – the red points have the same distance from each other as the blue points on every manifold. In the active decomposition (right), one of the manifolds encodes the circular translation of the triangle bump, while the second one represents the location of the square bump. Various colors indicate the location of the triangle. In the case of passive decomposition (left), since the transformation of individual shapes does not guide the decomposition, the manifolds jointly encode the location of each bump type.

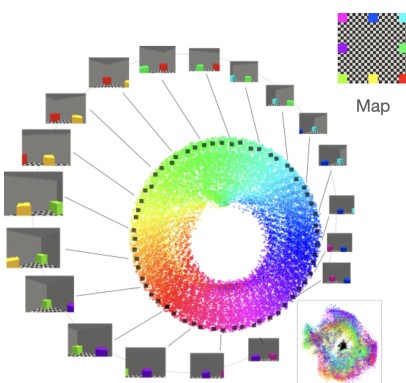

Figure 5: **Decomposition of the ego-motion.** The dataset contains a first-person view of a room. Transformations include right-left rotation and forward-backward movement. The equivariant embedding is produced by active decomposition using these two transformations, where the ring-structured manifold corresponds to the rotation action, and the smaller manifold corresponds to translations. Color coding shows the ground truth angle of the image. The black square markers show the traversal of the embedding as the agent rotates while standing in the middle of the room. Note that black squares are concentrated in the center of the second manifold.

observation consists of a first-person view of the room, downsampled to $32 \times 32$ pixels. The agent can rotate left/right or move forward/backward. We learn an $E(2) \times E(2)$ equivariant embedding using the active decomposition objective of Equation (5). Each mini-batch consists of 64 random observations and the result of applying all four actions in those states ($4 \times 64$ samples in total).

Figure 5 visualizes the embedding of the input in two sub-figures, where the more prominent figure shows the embedding corresponding to the rotation action, and the more petite figure (bottom right) shows the embedding corresponding to forward-backward movement. The first figure also shows the first-person view when the agent rotates while standing in the middle of the room. The corresponding markers collapse around the center of the second embedding, demonstrating an intuitive embedding parameterized by rotation angle and location. Walking straight across the room also produces the expected behavior of traversing the second manifold while the rotation angle, for the most part, remains fixed (not shown).

## 7.3 Quantitative Evaluation in Downstream Tasks

### 7.3.1 World Modelling

We select the Atari games Pong and Space Invaders as our environments for the world modeling experiments. These environments were previously used by Kipf et al. [31] to evaluate the Contrastive

| Environment | Method | H@1 | MRR |
|---|---|---|---|
| Atari Pong | World Model(AE) | $23.8_{\pm 3.3}$ | $44.7_{\pm 2.4}$ |
| | World Model(VAE) | $1.0_{\pm 0.0}$ | $5.1_{\pm 0.1}$ |
| | C-SWM | $36.5_{\pm 5.6}$ | $56.2_{\pm 6.2}$ |
| | **Ours** | $\mathbf{45.2}_{\pm 3.4}$ | $\mathbf{60.2}_{\pm 3.9}$ |
| Space Invaders | World Model(AE) | $40.2_{\pm 3.3}$ | $59.6_{\pm 3.5}$ |
| | World Model(VAE) | $1.0_{\pm 5.3}$ | $5.3_{\pm 0.1}$ |
| | C-SWM | $48.5_{\pm 7.0}$ | $66.1_{\pm 6.6}$ |
| | **Ours** | $\mathbf{54.2}_{\pm 6.3}$ | $\mathbf{68.7}_{\pm 5.1}$ |

Table 1: Hits at Rank 1 (H@1) and Mean Reciprocal Rank (MRR) of different methods.

| Methods | Inverted Pendulum | Reacher | Swimmer |
|---|---|---|---|
| Vanilla | $500_{\pm 150}$ | $-11_{\pm 2.5}$ | $25.6_{\pm 3.4}$ |
| AE-decoupled | $30_{\pm 15}$ | $-13_{\pm 3.0}$ | $16_{\pm 3.9}$ |
| AE-finetuned | $580_{\pm 130}$ | $-11.5_{\pm 3.2}$ | $26_{\pm 4.3}$ |
| In-SSL-decoupled | $100_{\pm 17}$ | $-15_{\pm 2.6}$ | $12_{\pm 2.5}$ |
| In-SSL-finetuned | $550_{\pm 21}$ | $-12_{\pm 4.1}$ | $25.9_{\pm 4.8}$ |
| Eq-SSL-decoupled | $456_{\pm 190}$ | $-14.8_{\pm 3.1}$ | $18_{\pm 4.5}$ |
| Eq-SSL-finetuned | $710_{\pm 120}$ | $\mathbf{-10}_{\pm 2.6}$ | $27_{\pm 3.5}$ |
| SymReg-decoupled | $800_{\pm 180}$ | $-14.5_{\pm 3.1}$ | $21_{\pm 4.1}$ |
| Dec-SymReg-decoupled | $600_{\pm 200}$ | $-12.8_{\pm 2.7}$ | $19_{\pm 5.6}$ |
| SymReg-finetuned | $\mathbf{950}_{\pm 50}$ | $-10_{\pm 3.4}$ | $\mathbf{31.5}_{\pm 3.9}$ |

Table 2: Average reward collected over 10 episodes for various models in Inverted Pendulum, Reacher and Swimmer. We provide the standard errors using 5 random seeds.

Structured World Model (C-SWM). We train the encoder using Euclidean SymReg of Equation (4), freeze it, and then learn a Multi-Layer Perceptron (MLP) based transition function in the latent space. Following Kipf et al. [31], we report Hits at Rank 1 (H@1) and Mean Reciprocal Rank (MRR), which are invariant to the embedding scale. These evaluation metrics measure the relative closeness of the following state's representation predicted by the transition model and the representation of the observed next state. We use a set of reference state representations to measure the relative closeness (embedding random observations from the experience buffer). Section 7.3.2 reports these measures and shows that a simple transition model learned on top of our embedding outperforms C-SWM in both games. Other reported baselines use an AutoEncodcer (AE) and a Variational AutoEncoder (VAE) to learn embeddings.

### 7.3.2 Reinforcement Learning

Next, we consider three Mujoco environments: InvertedPendulum, Reacher, and Swimmer from OpenAI Gym [5] and learn directly from the image observations. We compare our model with Auto-Encoder (AE) and Self-supervised Learning (SSL) based baselines. While AE learns to reconstruct the image observations of the states, SSL learns to inject invariance (IN-SSL) or equivariance (EQ-SSL) to agent actions. Given a triplet $(s, a, s)$, IN-SSL maximizes the likelihood of $f(s)$ and $f(s')$ being similar (SimCLR [10]). EQ-SSL of Dangovski et al. [18], in this context, additionally predicts the action that leads to the state transition. We introduce two variations of each model. In the first variation, the low-dimensional embedding is used as a substitute for the high-dimensional input data without further adjustment (-decoupled). The second variation allows for fine-tuning during the reinforcement learning stage (-fine-tuned). We use random policy to collect trajectories for the pre-training and use Proximal Policy Optimization (PPO) [49] algorithm for the downstream RL task. To evaluate the data efficiency of these models, we report the average reward collected over 10 episodes in the first 100,000 steps for Reacher and Swimmer and 30,000 steps for Inverted Pendulum in Section 7.3.2 (since Inverted Pendulum generally learns faster, we took a fewer number of steps.)

We see that out of all the representation learning methods, learned representations of SymReg most adequately capture the structure of the environment in Inverted Pendulum since the RL agent just trained on the fixed representation (SymReg-decoupled) outperforms all of them, including vanilla PPO. In Reacher, SymReg, along with other non-generative models, performs poorly compared to the AE. We believe that this is because the representation is focused on transformations caused by the agent's actions while details that can be valuable from the reward's perspective — in this case, the small object that the Reacher should reach - are ignored. This observation points to a limitation of all non-generative approaches that fine-tuning can resolve. To further verify this, we combined SymReg with a Decoder (Dec-SymReg) and noticed a significant improvement in the performance of the decoupled variation. In Swimmer, again, we see that learning the agent's transformations is not enough to get all the reward information as the background movement decides how far the agent has swum. Indeed, allowing the encoder to fine-tune allows the representations to reflect the reward information and improve performance.

## Conclusion

We proposed to learn equivariant representations by learning an injective embedding that is regularized towards a simple linear action using group invariants. We demonstrate this to be a simple, intuitive, and yet effective approach for representation learning. In the future, we would like to understand data characteristics that motivate the choice of one Lie group over others. We would also like to explore SymReg for the symmetric group and its combination with Euclidean groups as a way to represent various objects in Euclidean space. We would also like to investigate further the best choice of objectives or mechanism for preventing a representation collapse in conjunction with SymReg.

## Acknowledgments

We would like to thank anonymous reviewers for their constructive feedback. This research was in part supported by CIFAR AI chairs program and NSERC Discovery grant. The computational resources were provided by Mila and Compute Canada (now Digital Research Alliance of Canada). As a part of Mila, the authors acknowledge the material support of NVIDIA in the form of computational resources and IDT team and their technical support for maintaining the Mila compute clusters.

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
