# OpenReview forum: "Structuring Representations Using Group Invariants"
_NeurIPS.cc/2022/Conference — NeurIPS 2022 Accept_

### Official Review · Reviewer_oRnK · 2022-07-04

**Rating:** 6
**Confidence:** 3
**Soundness:** 3 good
**Presentation:** 3 good
**Contribution:** 3 good

**Summary:**

The paper proposes an approach that learns geometric invariants in the embedding space instead of in the input space. The authors point out that since injective functions are equivariant to any group, equivariant networks’ strong performance is a result of simple group actions. The proposed method does not constraint network structures and allows nonlinear or unknown group action on the input space. Experiments show that incorporating the geometric invariant into the loss function enables the neural network to learn meaningful embeddings and allows for decomposing of the symmetry group. Additionally, using the learnt embedding on downstream tasks achieves state-of-the-art performance.

**Questions:**

- For the world modeling experiment, why did the authors select the two particular experiments? It would help to provide justification that strong performance on the two games indicates that SymReg would perform well on the grid and 3-body physics environment too.
- From the argument in section 4, it seems that equivariance is still a useful information for non-injective functions such as many neural networks. When the group action is known, will SymReg outperform traditional equivariant models?

**Limitations:**

Limitations are listed in the conclusion. The authors also provide further discussion of some limitations (e.g. finite groups) in the appendix.

**Strengths And Weaknesses:**

Strengths:
- Inducing equivariance on the embedding instead of the input is novel. The observation that equivariance can be a weak inductive bias is insightful and the proposed method is well motivate. The distinction from previous works is clearly described in the related work section.
- The visualizations are very informative and provides strong support for the meaningfulness of the learnt embedding.
- The experiments are comprehensive, and empirical results support the advantage of the proposed approach compared to baselines.

Weaknesses:
- Although the paper addresses the problem of complex or unknown group actions in the input space, it still requires knowledge about which group to use on the embedding space. All experiments in the paper are performed on datasets that have physically meaningful symmetries. However, for tasks that do not have an obvious symmetry, it is unclear how to select a group.

Small typos: line 136 “X x G” should be “G x X” to be consistent with other places; Figure 3 caption “againts”

---

> ### Author Response · Authors · 2022-08-02
> **Response to Reviewer oRnK**
>
> We thank the reviewer for their constructive feedback, and we are happy to see their positive assessment of our paper. Below, we have tried to address all the questions or concerns raised in the review, and have updated the manuscript accordingly. In particular, we now have a discussion on defining symmetry regularization for a large family of “reductive groups,” and make practical suggestions on the choice of the group.
>
> ---
>
> __Q__: *“However, for tasks that do not have an obvious symmetry, it is unclear how to select a group.”*
>
> __A__: Since all Lie groups can be embedded in the Euclidean space of sufficiently large dimensions, a workaround is to use the Euclidean group as a default choice. Our 3D chair example supports this idea, where we recover the SO(3) manifold by learning an E(16)-equivariant embedding. Also, in our RL experiments, for environments without obvious symmetry, we demonstrate that using E(n)-equivariant embedding can lead to good representations which can capture the structure of transformation resulting from the agent’s action.
>
> ---
>
> __Q__: *“When the group action is known, will SymReg outperform traditional equivariant models?”*
>
> __A__:  Interesting point! First, note that in traditional methods, equivariance is guaranteed by the architecture, while here, it is guaranteed due to injectivity. Our SymReg loss encourages a simple group action on the latent space and is not guaranteed to vanish, therefore preventing a meaningful comparison. That said, we generally expect traditional methods to perform better in terms of generalization when dealing with fixed linear actions because the group action on the input is known, which should serve as a stronger inductive bias.
>
> ---
>
> __Q__: *“For the world modeling experiment, why did the authors select the two particular experiments? It would help to provide justification that strong performance on the two games indicates that SymReg would perform well on the grid and 3-body physics environment too.”*
>
> __A__: We chose the Atari experiments because, as suggested by results in C-SWM [10], they are the more challenging experiments. Moreover, most of the qualitative experiments in our paper were in a more symmetric setting similar to a 3-body physics environment or grid where we have already shown the efficacy of SymReg. If despite this rationale, the reviewer feels strongly about having these experiments, we can run them for the final draft.

---

### Official Review · Reviewer_tJ3i · 2022-07-08

**Rating:** 5
**Confidence:** 3
**Soundness:** 3 good
**Presentation:** 4 excellent
**Contribution:** 2 fair

**Summary:**

This paper discusses a self-supervised framework to learn equivariant representations (for a given group) from data. One of the key observations in this paper is that in most existing equivariant architectures, the group action on the embedding space is linear and hence limits the complexity of the architectures. The authors, therefore, propose to allow non-linear group actions and derive a simple relation for any function to be equivariant (Eq 2). The authors then translate this formula into a general recipe and an associated loss function (SymReg) that “regularises” for equivariance. More concretely, the authors translate this to \emph{preserve} geometric invariants of the respective group action. For eg, euclidean distances for SE(3), angles for conformal transformations, etc.

3 synthetic (Pendulum, chair, double bump world) and 2 real-world applications are shown. The main message in the demonstrations is that the embeddings learned by the proposed self-supervised framework are directly correlated to the actual variables of the manifold (for eg, angle, angular momentum, 3D Euler angles etc)

**Questions:**

Questions and Comments

- It is unclear how to deduce equation 4 from equation 2.

- More importantly, how does plainly minimizing equation 4 not descend into something trivial (like all 0s or constant). In the terminology of self-supervised learning, you have positives and negatives. If I interpret equation 4 as a positive, where is the negative? Or is SymReg used in conjunction with another loss? (it doesn’t appear so at least for the synthetic examples). To me, this is a very significant issue and I will wait for the discussion to make an opinion on this before labeling it as a strong drawback.

- (Suggestion) I would recommend more direct and compelling examples from 2D or 3D geometry. For eg, please see - (1.) Pai, Gautam, Aaron Wetzler, and Ron Kimmel. "Learning Invariant Representations Of Planar Curves." (2016). (2.) Basri, Ronen, and David Jacobs. "Efficient representation of low-dimensional manifolds using deep networks." arXiv preprint arXiv:1602.04723 (2016).





**Ethics Review Area:**

["I don’t know"]

**Limitations:**

See weaknesses. As a pre-rebuttal/discussion rating I am inclined to weigh in positively on this paper - for its clarity of discussion and a very interesting message on equivariant actions. However, there is still confusion in some aspects and I will wait for the discussion phase to make a more informed opinion.


**Strengths And Weaknesses:**

Strengths

- I found the articulation and clarity of “Equivariance is cheap actions matter” to be very nice and a good take home message
- Overall, this paper reads extremely well, and I appreciate the clarity of thought in most commentaries
- I also like the synthetic demonstrations, (~ sort of) validating the main claims of this paper (although I have some questions below)


Weaknesses


- Somehow, despite the main message of re-looking at equivariance through non-linear actions, I don’t get the impression at the end of the paper that the problem is solved. For eg, I see no demonstration (experimental, visual or otherwise) of the serious drawbacks of enforcing equivariance through linear actions and the benefits of allowing for non-linear group actions in practice. For eg, it would be a convincing demonstration if - the embeddings from some linear-action equivariant auto encoder are compared to the proposed regularisation and indeed there is a positive difference.

- One confusion that arises from this paper is: in the end what exactly is proposed? Is SymReg a manifold learning paradigm? (that generates more interpretable features?). In this case, it would be a good idea to also show the output of a similar dimensionality reduction scheme like [26].

- For the synthetic examples, the comparative demonstrations are pretty weak. For one thing, the size of the embeddings in figures 4 and 6 is much too small to be comfortable. it would really be more convincing to see the application of [26] for the experiment in figure 2 side-by-side with the proposed method.

- Some important issues, which I will phrase as questions.

---

> ### Author Response · Authors · 2022-08-02
> **Response to Reviewer tJ3i**
>
> We thank the reviewer for their detailed feedback and are happy to see their positive assessment of several aspects of our contribution. Below we address all the questions raised in the review; we have also updated the paper to accommodate the feedback and added the suggested experiment comparing our method to [26] in the appendix.
>
> ---
>
> __Q__: *“no demonstration (experimental, visual or otherwise) of the serious drawbacks of enforcing equivariance through linear actions and the benefits of allowing for non-linear group actions in practice.”*
>
> __A__: Note that we don’t claim that SymReg provides a better inductive bias compared to existing equivariant architectures; on the contrary, enforcing equivariance through architecture uses a stronger prior since it assumes the knowledge of group action on the input. The main limitation of equivariant architectures is their applicability to non-linear and unknown actions – e.g., we simply cannot apply existing equivariant architectures that use linear action to the mountain car or 3D chairs, where the transformations are non-linear.
>
> ---
>
> __Q__: *“what exactly is proposed? Is SymReg a manifold learning paradigm?”*
>
> __A__: Recent representation learning techniques using self-supervision can be seen as “amortized manifold learning” algorithms, where a neural network learns to perform the embedding. In contrast, most earlier manifold learning algorithms directly optimized the embedding. This is analogous to the comparison between traditional form variational inference, where the posterior is directly optimized for each input, and amortized inference, where a neural network produces the posterior. The machinery that distinguishes SymReg from old manifold learning techniques (in addition to amortization) is the central role of transformations for learning the embedding. In contrast, manifold learning methods often rely on a notion of distance or neighborhood, which is preserved in the embedding.
>
> ---
>
> __Q__: *“For one thing, the size of the embeddings in figures 4 and 6 is much too small to be comfortable. It would really be more convincing to see the application of [26] for the experiment in figure 2 side-by-side with the proposed method.”*
>
> __A__: We have increased the size of Figures 4 and 6. We have performed the suggestion experiment and added the result of applying DrLIM [26] to the paper; see Appendix D.1 for the experimental setup and results. Note that DrLIM [26] is similar to our SimCLR [10] implementation, with the difference being in the choice of contrastive loss. While SimCLR uses negative log-likelihood and cosine similarity, DrLIM uses hinge loss and Euclidean distance to achieve the pull (positive samples) and push (negative samples) behavior in the latent. We found that while SimCLR works reasonably in representation learning tasks (see Section 7.3.2) it fails to learn equivariance as shown in several examples in the paper. Our DrLIM experiments for rotating chair shows that, in general, contrastive learning methods are not very effective in learning equivariance.
>
> ---
>
> __Q__: *“It is unclear how to deduce equation 4 from equation 2.”*
>
> __A__: We show that equivariance of Equation (2) is enforced simply by having an injective encoder. Equation (4) is regularizing the group action on the latent space to make it close to a simple linear action. In the general setting, we achieve this regularization using the invariant polynomial bases for a given group. For more discussion on the general case, please see the newly added Section 5.1.
>
> ---
>
> __Q__: *“More importantly, how does plainly minimizing equation 4 not descend into something trivial (like all 0s or constant). In the terminology of self-supervised learning, you have positives and negatives. If I interpret equation 4 as a positive, where is the negative? Or is SymReg used in conjunction with another loss? (it doesn’t appear so at least for the synthetic examples). To me, this is a very significant issue and I will wait for the discussion to make an opinion on this before labeling it as a strong drawback.”*
>
> __A__: Collapse is prevented by the encoder being injective. As discussed in Section 4, having an injective encoder is sufficient for equivariance. However, equivariance only becomes a useful inductive bias when the group action on the embedding space is “simple.” Therefore, we need to ensure injective through some mechanism (e.g., a decoder, hinge loss), and use SymReg objective to regularize this injective embedding towards a simple linear action on the embedding. As stated at the end of Section 4, we use the logarithmic barrier function to avoid collapse. To make things more clear, we have added the detailed training algorithm of SymReg in Appendix C of the paper.

---

### Official Review · Reviewer_E8cu · 2022-07-11

**Rating:** 7
**Confidence:** 4
**Soundness:** 4 excellent
**Presentation:** 4 excellent
**Contribution:** 3 good

**Summary:**

This paper provides a methodological approach that enforces equivariance to certain to certain groups using only special losses during optimization. Their method is in the context of self-supervised learning. In the sense that similar to transforming auto encoder, in most cases, they have images and their transformed pairs in the batch. Then they constrain the learned embedding by enforcing them to be in a certain space. They do so by enforcing the defining invariant of that space. For example for the case of Euclidean space, the defining invariant is the distance. So they optimize the fact that the distance of embeddings for two images should stay the same before and after a transformation.
There are known defining invariants for many groups. Most interestingly they can easily enforce equivariance to Affine transformations by simply forcing the embeddings to be euclidean.
They also provide modification that can handle decoupled groups by adding invariance to other groups for each data point of one group.

There is enough evidence in the experiments that suggests the work delivers what it promises. The visualizations specially show that being Euclidean for SO(3) equivariance makes it not only equivariant but easy to investigate and representations are intuitive. The practical application in RL show that when the equivariance is the key factor adding their loss strongly improves the result. When other factors such as background or details are also important adding a finetuning and generative decoding would relax the model and enables it to encode more aspects of the scene.

Two aspects that was most impressive were the fact that they only input the images and that there is no architectural constraints or special architectural implementations. The transformation can be even unknown (already exists in the data...) so it's not reliant to only transformations that can be done by our code during data pre-processing. Also, since it is architecture agnostic it can be applied on any network however it is implemented, specialized for a specific processor, etc.

One aspect that is concerning is that since it is only a matter of optimization, what happens if the loss is competing with other losses. For example if the network is trained for a specific task, such as segmentation, then one needs to balance the factors between the equivariant losses and the final loss. Since it is not architectural, it is not guaranteed that the embeddings are equivariant. If you lower the factor on the proposed loss compared to the task loss so that segmentation results get better, it could just simply not be equivariant and at the test time it won't generalize to new viewpoints.

**Questions:**

Where is the code? In the questionnaire you have answered yes to the code one, but I couldn't find the url and the zip only contains a pdf.

How do you project to lower dimensions  for visualization? In the decoupled case of the bumps with the 4 dimensions do the dimensions readily form the circles or do you do any projections?

**Ethics Review Area:**

["I don’t know"]

**Limitations:**

This work does not increase the negative social impact of the literature.

Whether their regularizer is applicable to setups with natural images is yet to be seen and it's their current limitation.

**Strengths And Weaknesses:**

Strengths: The proposed method is theoretically sound. They clearly state their math, ideas, and proofs. The manuscript is well written and easy to follow and understand. The idea is novel and applicable. They provide clear experiments showcasing the strengths of their method in the RL domain.

Weaknesses: The regularizer can be too strict (enforcing distance equivariance for example) for a downstream task. For the RL case they mitigate the issue by adding a generative decoder and finetuning. It is not clear if after finetuning the embedding is still equivariant or not.
Also, it is not shown that it is applicable in a natural image setup or tasks.

---

> ### Author Response · Authors · 2022-08-02
> **Response to Reviewer E8cu**
>
> We thank the reviewer for their detailed feedback and are happy to see their positive assessment of our contribution, presentation, and empirical evaluations. Below we address the questions; we have also updated the manuscript to accommodate the feedback.
>
> ---
>
> __Q__: *“since it is only a matter of optimization, what happens if the loss is competing with other losses.”*
>
> __A__: We show that having an injective encoder is sufficient for equivariance, and our objectives are simply “regularizing” the group action on the embedding space. Our SymReg objective should therefore be seen as a regularization, which can have its own hyperparameter without compromising the equivariance of the embedding.
>
> ---
>
> __Q__: *“For the RL case they mitigate the issue by adding a generative decoder and finetuning. It is not clear if after finetuning the embedding is still equivariant or not.”*
>
> __A__: Note that the representation remains equivariant as long as the encoder remains injective. However, we agree with the reviewer that a current limitation in the RL application is that our injectivity constraint is not enforced during the fine-tuning, which can undermine equivariance.
>
> ---
>
> __Q__: *“Where is the code?”*
>
> __A__: Thanks for the reminder; we have added the code to the supplementary material.
>
> ---
>
> __Q__: *“How do you project to lower dimensions for visualization? In the decoupled case of the bumps with the 4 dimensions, do the dimensions readily form the circles or do you do any projections?”*
>
> __A__: The projections for visualization are using “random” orthogonal matrices; For example, to project from 10D to 3D, we randomly pick three orthogonal vectors in the original space and project everything onto that basis. In the disentangled bump world example, there is no projection happening since we use $E(2) \times E(2)$ for the embedding space.

---

### Official Review · Reviewer_44UN · 2022-07-11

**Rating:** 7
**Confidence:** 2
**Soundness:** 3 good
**Presentation:** 2 fair
**Contribution:** 3 good

**Summary:**

The paper advocates the use of invariants for learning symmetry-preserving (equivariant) representations. Rather than building a network that maintains group actions at each stage of processing (as commonly done in approaches such as G-convolutional networks and their steerable descendants) abstractly or concretely, the idea here is to enforce the preservation of invariants as an objective of the learning process (for example, maintaining distances for E(d)). For factored groups, multiple embeddings can be used to capture actions of each factor group, and it is also possible to use less specific invariants to capture subgroups (for example, maintain SO(2)-covariance for groups of discrete rotations).

The paper applies this idea to a set of example applications and discusses the results and findings.

**Questions:**

- Is there a generic "algorithm" how to apply the ideas of this paper in practice, or is a case-by-case assessment needed to find invariants, objective functions and suitable networks to search for them?
- The idea of embeddings with invariants seems to be tightly related to metric learning and approaches such as generalized MDS (multi-dimensional scaling). Can one reuse ideas from that field?


**Limitations:**

As stated above, I see some limitations in terms of costs of computing the embedding and the ability to reach the required precision (depending on the application). I think this is worth discussing a bit more in the paper.

**Strengths And Weaknesses:**

I would like to start the discussion of this submission by stating that I found it difficult to understand what the paper is aiming. The observation of Section 4 that equivariance (homomorphism into some group representation on the output side) is almost trivial (only invertibility required) when general group actions (or representations of the group) are allowed is certainly valuable and seems to be a valuable call to caution in terms of how to formally characterize what is actually desired. The proposed solution is to identify invariants and enforce them using soft-constraints in the objective function.

The criticism of learning equivariant representations by using multiple stages of suitable group representations has targets two problems: First, the representation could theoretically be meaningless, but this does not happen in practice where specific representations are designed the structure of which is well-understood and can be used to reconstruct invariant embeddings. For example, steerable networks for use in areas such as computational physics aim at covariant computations to maintain conservation laws (for example, by learning covariant force-fields or potentials with covariant derivatives); here, the structure of the equivariant map is very important (although this aspect might not have been emphasized explicitly that much). Secondly, such representations might not be suitable for *discovering* symmetry in data, possibly encoded in a highly non-linear way. Here, the usage of invariance-preserving embeddings could indeed be beneficial, as less constraints on the form of the symmetry and group action are needed.

On the technical side, some problems remain though: The idea of finding embeddings with the same symmetry properties as the input is compelling in its simplicity and easy of use in down-stream applications, and the idea of setting up larger abstract groups the actual symmetry in the data only forms a subgroup of leads to some interesting applications. However, there are difficulties with this approach:
- Invariants enforce via an objective function are approximate and have to be traded-off with other potential constraints of the embedding (which might be quantified in a different unit of measurement, thereby complicating parameter choice for varying sampling densities or other settings).
- The invariants have to be derived manually; the paper only gives some examples of well-known cases. From the discussion, it is not clear (and I personally do not know) how to generally map between groups and their invariants.
- Even if the invariants are enforced as much as possible, the network itself might only provide approximate symmetry if not designed towards equivariance. For some applications (as physical simulations), accuracy is an essential requirement.
- Learning invariants might be very costly, as the unconstrainted network is much larger: For example, trying to learn the shift-invariance of a CNN by distance constraints to a fully-connected MLP would be very costly (theoretically doable, but not practical). Similar problems could easily occur in other cases such as fin-granular rotations, as the non-symmetric network needs sufficient representational capacity.

My impression is that the paper proposes a rather elementary mechanism that is hard to implement effectively (for some applications, where precision of symmetry matters) and efficiently (for applications with many degrees of freedom where symmetry is a strong constraint). It remains a bit vague on the implementation side, only showing a collection of examples. That said, some of the examples are quite interesting as they automatically discover relevant structure.

Overall, I am rather uncertain what to recommend. Many of the issues that I have perceived might be a problem of positioning and writing, but I do think that some non-trivial technical problems remain that are not easily solved (and that should be discussed). At the current point, I would give a negative overall assessment for NeurIPS 2022, but I would be happy to change my rating if the further discussion clarified misconceptions on my side.

**Update:** After reading the other reviews and the rebuttal (thanks to the authors for their detailed explanation) I understand the contribution better (non-linear vs. linear embeddings and why it is a good idea to just regularize the learned embedding vs. prescribing one). The issue of approximate regularization (which was also brought up by other reviewers) seems to remain (the method cannot guarantee exact invariants), but this is not a "show-stopper" (some applications rely on this, but many do not, and the gained ability to discover new symmetry seems very promising). I have therefore (in line with the other reviewers) updated my rating towards a positive recommendation.

---

> ### Author Response · Authors · 2022-08-02
> **Response to Reviewer 44UN**
>
> We thank the reviewer for their feedback. We have tried to address all questions raised in the review and have updated the paper to clarify these points further.
>
> ---
>
> __Q__: *“Invariants enforce via an objective function are approximate and have to be traded-off with other potential constraints of the embedding”*
>
> __A__: In section 4 we show that equivariance is guaranteed by having an injective map, and the effect of our symmetry objective is to “regularize” this embedding. As with any regularization, SymReg can be used with a hyper-parameter that controls the trade-off without compromising equivariance. Results in all of our experiments use a hyperparameter equal to one.
>
> ---
>
> __Q__: *“The invariants have to be derived manually”*
>
> __A__: The invariants for classical Lie groups and finite groups are well-known and studied under invariant theory. To further clarify the use of invariants in our work, we have added Section 5.1 with a brief discussion on invariant theory and the use of invariant bases. Note also that existing equivariant networks are arguably more specialized to particular groups in deriving their architecture.
>
> ---
>
> __Q__: *“Even if the invariants are enforced as much as possible, the network itself might only provide approximate symmetry if not designed towards equivariance.”*
>
> __A__: This is a misunderstanding. Our theorems show that having an injective encoder is sufficient for exact equivariance. However, equivariance, as an inductive bias, becomes useful with a simple group action on the latent space, and our symmetry objective regularizes the embedding towards this goal.
>
> ---
>
> __Q__: *“Learning invariants might be very costly, as the unconstrainted network is much larger.  For example, trying to learn the shift-invariance of a CNN by distance constraints to a fully-connected MLP would be very costly”*
>
> __A__: This may or may not be the case: first, note that existing equivariant architectures enforce equivariance on each layer; this is not strictly necessary and enforcing it on the entire network may lead to more expressive models. Moreover, the argument in the review assumes that the translation group is acting on the latent space through the permutation of axes (pixels). This does not have to be the case, and in our approach, one could act on the latent by translating the latent embedding, therefore producing a more compact representation using a smaller network.
>
> ---
>
> __Q__: *“Is there a generic "algorithm" how to apply the ideas of this paper in practice, or is a case-by-case assessment needed to find invariants, objective functions and suitable networks to search for them?”*
>
> __A__: We do require the specification of group invariants for our algorithm. However, since any group manifold can be embedded into the Euclidean space, choosing a large enough Euclidean space can work well in many applications. Our experiments also confirm this, where we show that our E(16)-equivariant embedding of the 3D chair is (consistent with) the SO(3) manifold. We have added further discussion on this (see Section 5.1) and a generic algorithm for E(n) (see Appendix C) to the revised manuscript.
>
> ---
>
> __Q__: *“The idea of embeddings with invariants seems to be tightly related to metric learning and approaches such as generalized MDS (multi-dimensional scaling). Can one reuse ideas from that field?”*
>
> __A__: There is a nuanced but crucial distinction between the idea of isometric embedding and distance preservation due to transformations. The former learns an embedding that preserves the distances in the input calculated on the data-manifold, while the latter, when using the Euclidean group, preserves the distances before and after a transformation, both in the latent space. We briefly discuss this distinction at the end of our related works section.

---

> > ### Author Response · Authors · 2022-08-09
> > **Followup on Response to Reviewer 44UN**
> >
> > As today is the last day of the discussion period, we would like to know if the reviewer needs any further clarifications on the paper or our comments.
> >
> > We will also like to reiterate that the goal of this work is to design a method to learn equivariance and regularize it for linear action in the latent using Invariants. This is particularly useful when the action of the symmetry group is not linear or unknown in the input space where we cannot apply existing equivariant architectures. Eg. As the observations are in a 2D pixel space in the Rotating Chair experiment we can’t use an E(2) equivariant CNN to learn an equivariant embedding space with linear action with respect to SO(3). Similarly in the RL setup without obvious symmetries using SymReg is a better way to incorporate equivariance with linear action in the latent. That said, we generally expect existing equivariant networks to perform better when dealing with fixed linear actions because the group action on the input is known, which should serve as a stronger inductive bias.

---

### Author Response · Authors · 2022-08-02
**General Response**

We thank all the reviewers for their thoughtful feedback, concisely outlining our paper's theoretical and empirical contributions. We will like to summarize the changes made in the revised manuscript (highlighted in blue):

1. We have added a new section on the use of invariant polynomial bases, which gives a general recipe for deriving symmetry regularization objectives for reductive groups.
2. Based on the reviewers’ constructive feedback, we have added algorithms and additional experiments in the appendix.
3. We did minor clarifications and fixed minor typos based on the feedback.

---

### Meta-Review · Area_Chair_zyiU · 2022-08-28

**Recommendation:** Accept
**Confidence:** Certain

**Metareview:**

This paper proposes a self-supervised framework for learning equivariant representations (for a given group) from data. Applications in reinforcement learning support the claims in the paper. All reviewers agreed that the approach is interesting to the neurips community. Accept

**Award:**

No

---

### Decision · Program_Chairs · 2022-09-14

Accept